# Additional Benefits of Creatine Supplementation with Physical Therapy and Resistance Exercise in Knee Osteoarthritis: A Randomized Controlled Trial

**DOI:** 10.3390/jcm14238538

**Published:** 2025-12-01

**Authors:** Muhammad Osama, Sabah Afridi, Bruno Bonnechère

**Affiliations:** 1Department of Physical Therapy, Ibadat International University Islamabad (IIUI), Islamabad 44000, Pakistan; 2Active Health Physio, Timaru 7910, New Zealand; 3Brainstorm Research, Islamabad 45710, Pakistan; sabahafridi@gmail.com; 4Isra Institute of Rehabilitation Sciences (IIRS), Isra University Islamabad Campus, Islamabad 44000, Pakistan; 5REVAL Rehabilitation Research Center, Faculty of Rehabilitation Sciences, Hasselt University, 3590 Diepenbeek, Belgium; bruno.bonnechere@uhasselt.be; 6Technology-Supported and Data-Driven Rehabilitation, Data Sciences Institute, Hasselt University, 3590 Diepenbeek, Belgium; 7Department of PXL—Healthcare, PXL University of Applied Sciences and Arts, 3500 Hasselt, Belgium

**Keywords:** creatine, knee osteoarthritis, physical therapy, physiotherapy, resistance exercise, strength training

## Abstract

**Background**: Knee osteoarthritis (KOA) is a progressive joint disorder that leads to pain, functional limitations, and reduced quality of life. While physical therapy (PT) and resistance exercise are effective in managing KOA, creatine supplementation (CS) may provide additional benefits. **Aims/Objectives**: To determine the additive effects of creatine supplementation alongside physical therapy (PT) and resistance exercise training in individuals with KOA. **Methods**: A parallel-design, double-blind, randomized controlled trial was conducted on 40 patients with KOA (≤grade III on Kellgren classification), aged 40–70 years. Participants were randomly allocated to either a placebo control group, which received placebo supplementation (maltodextrin) along with PT, including heat therapy, electrotherapy, manual therapy, and resistance exercises, for four weeks, or an experimental group, which received CS instead of maltodextrin in addition to the same treatment. Outcome measures included the visual analog scale (VAS), fall risk, Knee Injury and Osteoarthritis Outcome Score (KOOS), isometric muscle strength (IMS), five-repetition sit-to-stand test (5xSST), knee range of motion (ROM), and body composition analysis. **Results**: No significant differences were observed between the two groups at baseline. After four weeks of treatment, a significant interaction effect (treatment group x time) was observed for VAS (*p* = 0.001), fall risk score (*p* < 0.001), KOOS overall score (*p* < 0.001), IMS (*p* < 0.001), and body composition parameters (*p* < 0.05) in favor of the CS group. However, no significant interaction effect was observed for knee ROM and KOOS QOL subscale. **Conclusions**: CS, when combined with PT and resistance exercise, may provide additional benefits in terms of pain, function, muscle strength, and body composition parameters in individuals with KOA. However, no supplementary benefits of CS are observed in terms of quality of life and ROM.

## 1. Introduction

Osteoarthritis (OA) is the most common degenerative joint disorder and chronic rheumatic disease affecting 7% of the population worldwide [1,2,3,4,5]. Although, any synovial joint can be affected by OA, an estimated 250 million people suffer from knee OA (KOA), making the knee the most common site of symptoms [1]. KOA is associated with pain, stiffness, loss of function, and walking-related performance fatigability [6,7], and is the 10th leading cause of disability worldwide [5]. The prevalence of radiographic KOA is observed to be 31% in men and 34% in women aged from 63 to 94 years [1]. On the other hand, the prevalence of symptomatic KOA is observed to be 6.8% and 11.4% in men and women, respectively [1]. Even though the prevalence of OA has increased by 48% from 1990 to 2019, mostly due to the aging population, the advancements in OA treatment have not been remarkable [3,4].

To date, there has been no known cure for OA, and pharmacological treatment has been the mainstay of conservative management, but their prolonged use can have numerous side effects, including opioid use disorder [3,8]. This has led to an increased emphasis on the conservative non-pharmacological management of KOA, including dietary supplements, physical therapy, and exercise [1,3,9,10,11,12,13]. Even though supplements like glucosamine and chondroitin sulfate are commonly used for the management of KOA, evidence regarding their effectiveness is inconclusive [14,15]. In particular, when combined with physical therapy and exercise, these supplements seem to offer no additional benefit [14,15]. In terms of therapeutic exercise, resistance training is more suited for the management of KOA as it involves greater load but fewer repetitions, allowing greater gains in strength and performance without causing a lot of repetitive movement between the two degenerated joint surfaces, and it has been shown to exert positive effects on clinical and physiological outcomes [12,16,17,18,19]. Research has also shown loss of muscle strength and quadriceps femoris weakness being a significant contributor to knee pain and related disability in KOA [16,20,21], with 11–56% and 76% lower concentric and eccentric extensor strength, respectively [22]. These findings advocate the use of resistance training in persons with KOA, as it is superior to aerobic training and general exercises in terms of improving muscular strength and size [23,24].

The effects of exercise training can be augmented by special diet or external supplementation, and previous research has shown that creatine is one of the most common and effective supplement in sports and rehabilitation to augment resistance training for improving muscular strength and performance [12,25]. Creatine is synthesized from amino acids and is stored in the skeletal muscles as free creatine and phosphocreatine (PCr) [12,25,26]. Within muscle tissue, creatine supports rapid energy regeneration, as PCr serves as an immediate energy reserve, allowing quick ATP re-synthesis during short, high-intensity activities such as resistance exercise and strength training. By increasing PCr availability, creatine supplementation enhances ATP production, enabling greater force generation and increased exercise volume, capacity, and intensity. Over time, this contributes to enhanced muscle strength and hypertrophy [12,25,26]. Furthermore, creatine supplementation supports muscle repair and growth by stimulating satellite cell signaling and increasing intracellular water (ICW), leading to muscle cell expansion [12,25,26]. It may also reduce muscle breakdown by lowering myostatin levels, thus supporting lean mass development. Moreover, studies show that creatine supplementation facilitates recovery, prevents injury, and displays neuro-protective and anti-inflammatory effects [12,25,26]. The potential therapeutic benefits of creatine supplementation have been reported in several conditions such as muscular dystrophy, Parkinson’s disease, stroke, and even in aging and pregnancy [12,25,26]. Moreover, creatine supplementation in combination with resistance exercise training has been observed to be safe and effective in improving skeletal and muscular health in older adults [12,27]. However, there is scarcity in the literature regarding the effects of creatine supplementation in the management of KOA, and for this reason, the purpose of the current study was to determine the additive benefits of creatine supplementation, when combined with resistance exercise training and physical therapy in the management of KOA.

## 2. Materials and Methods

### 2.1. Design and Setting

A parallel-design, double-blind, randomized controlled trial was conducted at Foundation University College of Physical Therapy (FUCP) from June 2020 to December 2023, in which a total of 40 participants were included (Figure 1). Informed consent was taken from all participants before inclusion in the study, and participant confidentiality was maintained. Ethical approval was taken from the Advance Study and Research Committee, Isra Institute of Rehabilitation Sciences (Ref#1809-PhD-004), and from Foundation University Islamabad (Ref#FF/FUMC/215-30/Phy/20), and the study was prospectively registered at clinicaltrials.gov (NCT04423887). Informed written consent was acquired from all participants, and the participants had the right to withdraw from the study at any point. As the study involved a short-term, low-risk intervention consisting of supervised physical therapy and creatine supplementation within safe, evidence-based dosages, no formal data monitoring committee was formed. Participants were, nevertheless, monitored throughout the intervention period for any adverse events. It is important, however, to note that no adverse events were reported.

Patients, referred by a physiatrist to the FUCP Multi-Disciplinary Lab, were recruited in the study and were selected via purposive sampling, after which they were randomly allocated to the control (placebo) group and experimental (creatine supplementation) group. Randomization was carried out using a simple randomization (coin toss) method by an independent researcher not involved in data collection or assessment. Group allocation (creatine supplementation or placebo) was concealed using opaque, sealed envelopes opened only after participant enrollment to ensure allocation concealment. Both participants and assessors were blinded to the treatment group allocation to minimize bias. The treatment was given to both groups for 4 weeks.

### 2.2. Eligibility Criteria

Patients aged from 40 to 70 years, having a history of KOA no less than 3 months, knee pain no less than 4/10 and no greater than 8/10 on visual numeric pain rating scale (VNRS) during the last week, and radiological evidence of grade III or lower on the Kellgren classification, were included in the study. Patients were referred by a physiatrist and had a confirmed diagnosis of KOA based on radiographic findings. The X-ray reports were issued and graded by a radiologist and included the severity grade of KOA. Patients with signs of serious pathology such as malignancy, inflammatory disorder or infection; a history of trauma or fractures in the lower extremity; signs of lumbar radiculopathy or myelopathy; a history of knee surgery or replacement; and/or receiving intra-articular steroid therapy, creatine, or joint supplements in the last two months were excluded.

### 2.3. Study Interventions

#### 2.3.1. Conventional Physical Therapy

Physical therapy intervention was administered 3 times a week (1 h sessions), for 4 weeks, and a combination of superficial heating and two pole interferential therapy (Intelect® Advanced Therapy Combo Unit, Chattanooga®, Dallas, TX, USA) with a carrier frequency of 2500 Hz for 20 min was used for pain modulation. This was followed by joint mobilization, consisting of tibio-femoral anterior and posterior glide and patello-femoral joint mobilization (Figure 2). All treatments were administered by the same physical therapist.

#### 2.3.2. Resistance Exercise Training

Resistance exercise training was carried out as supervised exercise training three times a week and as home exercise program for the remaining 4 days, which was ensured via diary keeping. The exercise session was started with 10 min of pain-free, self-paced walking for warm-up, followed by resistance exercise training, including leg press, concentric and isometric knee extension and flexion in sitting, sit-to-stand (mini squats), and stationary cycling with maximum resistance, as per tolerance till failure [16]. Three sets of 8 repetitions were carried out for each individual exercise, allowing 10 to 15 s rest between repetitions and 1 to 2 min rest between sets [16,18]. In addition, terminal knee extension was also performed for improving extension lag. For equipment-based supervised resistance training, 80% of 8 repetition maximum was used as training intensity, which was reassessed every week [16,18]. Treatment intensity was reduced in case of increase in pain or inability of the patient to perform the exercise. If the exercise still caused pain, the specific exercise was halted and skipped for that session. Home exercises consisted of exercises which can be performed easily by the participants at home without any specialized equipment, including self-paced walking, concentric (with weight cuffs) and isometric (against manual resistance with the help of a family member) knee extension and flexion, terminal knee extension, and sit-to-stand (mini squats) with and without weight, as per tolerance till failure so that the exercises do not exacerbate pain. Pictorial representation of exercises is included in Figure 2 and Figure 3.

#### 2.3.3. Creatine Monohydrate Supplementation

The participants in the creatine supplementation group received creatine monohydrate supplementation (The Protein Works ^TM^, Runcorn, Cheshire, UK) of 20 g per day during the first week for creatine loading, followed by 5 g per day for the remaining 3 weeks until the study was concluded, which is in line with the dosage and loading protocol recommended by the International Society of Sports Nutrition [12,25,26]. In the first week, the participants were given servings of 5 g, four times each day, instead of one serving of 20 g. The control group received a placebo (maltodextrin packets) in the same pattern to ensure blinding. Maltodextrin was used as the placebo as it closely matches the appearance, texture, and solubility of creatine monohydrate, ensuring effective blinding without producing any physiological effects at the administered dose. Moreover, several previous studies investigating CS have also employed maltodextrin as a placebo for the control groups [12]. Participants were provided with pre-weighed, individually labeled sachets of creatine monohydrate (or placebo) for daily consumption, and were instructed to record their intake in a supplementation and exercise log diary, which was reviewed weekly by the research team during supervised physical therapy sessions.

Regarding tolerability, the loading dose (20 g/day divided into four 5 g doses), followed by a maintenance dose (5 g/day), is consistent with the standard guidelines of the International Society of Sports Nutrition and has been shown to be well tolerated, with minimal gastrointestinal effects [12,25,26]. Furthermore, no participants reported adverse gastrointestinal symptoms or noticeable water retention in the current study, suggesting that unblinding was unlikely.

### 2.4. Outcomes

#### 2.4.1. Pain, Symptoms, Function and Quality of Life

The visual analog scale (VAS) and the Knee Injury and Osteoarthritis Outcome Score (KOOS) were used as the main patient-reported outcome measures. VAS was used to quantify pain which has an intra-class correlation coefficient (ICC) of 0.97 for test–retest reliability for KOA [28]. Moreover, KOOS, a self-administered questionnaire was used to quantify pain, symptoms, activities of daily living (ADL), sports and recreation function, and quality of life (QOL), and has an ICC of 0.83 to 0.89 for KOA [29]. Each item was scored on a 5-point Likert scale, and the raw scores were transformed to a 0–100 scale, where a higher score signifies a better outcome.

#### 2.4.2. Muscle Strength

Isometric muscle strength of the knee flexors and extensors was assessed using a modified sphygmomanometer test, following procedures adapted from Silva BBC et al. [30]. Participants were instructed to sit upright, with hips and knees flexed to 90°, feet unsupported, and arms crossed. A stabilization belt was applied around the distal third of the leg and bed. For knee flexion, the sphygmomanometer cuff was placed on the posterior aspect of the distal part of the leg against the bed, and for extension, on the anterior aspect of the distal part of the leg against the belt. After a familiarization trial, participants performed one maximal 5 s isometric contraction, cued verbally, and instructed to avoid the Valsalva maneuver. The peak pressure (mmHg) was recorded as the isometric muscle strength value. The procedure is shown to have high inter-rater and intra-rater reliability for both knee flexors and extensors, with an ICC > 0.83 [30].

Furthermore, the five-repetition sit-to-stand test was also used to analyze muscle strength, as it has a statistically significant correlation with strength in knee extension [31], with high ICC for both inter-observer and test–retest reliability (0.98 and 0.99, respectively) [32]. The participants were instructed to stand up from a sitting position followed by sitting, and repeating the procedure five times as fast as they can [33].

#### 2.4.3. Range of Motion

Knee flexion and extension range of motion was quantified using goniometry, which has been shown to have high reliability and narrow limits of agreement for patients with KOA, with an ICC of 0.96 [34].

#### 2.4.4. Body Composition

Body composition analysis was carried out using multi-frequency direct-segmental bio-electrical impedance analysis (InBody Co., Ltd., Gangnam-gu, Seoul, Republic of Korea) using eight tactile electrodes [35], with high intra-class correlation coefficients (>0.80) and narrow limits of agreement for lower extremity lean muscle mass, body lean mass, body fat percentage, percentage body weight, and total body water when compared with dual-energy X-ray absorptiometry (DEXA) as the reference standard [35,36]. The test was carried out indoors, with regular clothes, standing barefoot on the foot of the electrodes, and with arms slightly abducted with hands grasping the hand electrodes [35]. Phase angle, intracellular water (ICW) ratio for the whole body as well as both lower extremities, visceral fat area, percentage body fat, skeletal muscle mass, segmental lean mass and segmental fat mass for both lower extremities were calculated via bio-electrical impedance analysis and analyzed as outcome measures. To minimize variability, all measurements were standardized, as participants were instructed to avoid food intake, exercise, and caffeine for at least 4 h before assessment, and testing was performed at a similar time of day for all participants.

#### 2.4.5. Fall Risk

Fall risk was assessed using the Biodex Balance System (Biodex Medical Systems Inc., Shirley, NY, USA), which is shown to have a test–retest reliability ICC of 0.64 to 0.91 [37] and has been used previously in musculoskeletal conditions, including KOA, to evaluate balance impairments and fall risk [38,39,40]. For the fall risk assessment, participants were instructed to stand with their eyes open on an unstable platform set at stability level 6. Each participant completed three 20 s trials, the average of which was computed afterwards. Before the test, a familiarization trial was conducted to ensure that participants understood the procedure, and those scores were not recorded. During the test, participants were instructed to maintain their balance. A higher score on the Biodex Balance System is indicative of a greater risk of falling [39,41].

### 2.5. Statistical Analysis

A sample size estimation was performed prior to the experiments using the OpenEpi sample size calculator [42]. A total sample of 34 patients was calculated using the data from Peeler J et al. [43] for VAS, with a mean 2.17 ± 1.95 for the control group and 0.7 ± 0.85 for the experimental group, confidence interval of 95%, and a power of 80%. A total of 40 participants (20 per group) were included in the study to compensate for any potential drop-out or loss in follow-up. VAS was selected as the primary outcome measure because pain reduction was the principal clinical endpoint of interest and had the most robust prior data available for sample size estimation. Given the limited research on CS in KOA, and lack of reliable mean difference data for other outcomes were unavailable, additional outcomes were therefore exploratory, and the present study provides valuable preliminary data for future power calculations in this emerging research area.

Normality of the data was checked using the graphical method (boxplot and QQ-plot). Descriptive statistics were reported in the form of mean ± standard deviation (S.D), and mean differences were reported, with 95% confidence intervals. Fischer’s exact test was used to compare the two groups in terms of categorical variables, i.e., gender and grade of knee OA. Independent *t*-tests and paired t-tests were used for between-group and within-group comparisons. Furthermore, mixed ANOVA was used to compare the two groups with Bonferroni correction to determine the interaction effect (treatment group × time), main effect (time), and main effect (treatment group). In addition, partial eta squared (η^2^p) was reported as a measure of effect size to interpret the magnitude of these effects. All the statistical analysis was carried out using SPSS v 21.0, and a confidence interval of 95% was used, with a *p*-value of less than 0.05 considered significant.

For the analysis of the results of the current study, an intention-to-treat approach was considered. However, as no dropouts occurred, analyses were conducted on the full dataset of all randomized participants.

## 3. Results

### 3.1. Individual Characteristics

A total of 40 participants were included in the study, with 20 participants in each group. The two groups were found to be similar in terms of distribution of gender and grade of KOA (Table 1). Moreover, no dropouts were reported in the study (Figure 1). Furthermore, no statistically significant differences were observed between the two groups in terms of age, weight, height, and body mass index (BMI) (see Table 2).

### 3.2. Pain, Symptoms, Function, and QOL

No statistically significant differences were observed between the two groups in terms of VAS scores, KOOS overall scores, or any of the subscale scores at baseline (Table 3). However, a significant interaction effect (treatment group × time) was found between time and group for VAS scores [F(1,38) = 13.113, *p* = 0.001, η^2^p = 0.257], indicating that pain reduction over time differed between the control and experimental (creatine) groups, and even though both groups showed a significant reduction in VAS scores over time, creatine group demonstrated a greater reduction (see Figure 4). Similar results were reported for KOOS overall score (Table 3) and all of its subscales, except for QOL subscale, with no significant interaction effect [F(1,38) = 4.077, *p* = 0.051, η^2^p = 0.097] or main effect of the treatment group [F(1,38) = 0.502, *p* = 0.483, η^2^p = 0.013]; however, a significant main effect of time was reported [F(1,38) = 795.715, *p* = < 0.001, η^2^p = 0.954].

### 3.3. Fall Risk

No significant differences were observed between the two groups in terms of fall risk score (FRS) at baseline (*p* = 0.635). However, a significant interaction effect was found between time and group for FRS [F(1,38) = 28.381, *p* = <0.001, η^2^p = 0.428], indicating that improvement in FRS over time differed between the control and experimental (creatine) groups, and even though both groups showed a significant improvement in FRS over time, the creatine group demonstrated a greater improvement as shown in Table 3.

### 3.4. Range of Motion

No significant differences (*p* > 0.05) were observed between the two groups on the symptomatic or asymptomatic side, neither in knee flexion nor in knee extension ROM, at baseline. No significant interaction effect was found between time and group for ROM on both sides (Table 3), except for knee flexion ROM on the asymptomatic side [F(1,38) = 4.727, *p* = 0.036, η^2^p = 0.111], but even then, a significant main effect only for time was reported [F(1,38) = 200.239, *p* = <0.001, η^2^p = 0.840] but not for the treatment group [F(1,38) = 1.180, *p* = 0.284, η^2^p = 0.030]. Similarly, a significant main effect only for time was reported for knee flexion (symptomatic side) [F(1,38) = 1769.384, *p* = <0.001, η^2^p = 0.979], knee extension (symptomatic side) [F(1,38) = 631.184, *p* = <0.001, η^2^p = 0.943], and knee extension (asymptomatic side) [F(1,38) = 153.740, *p* = <0.001, η^2^p = 0.802]. These findings signify that even though there are no significant differences between the two groups, a significant improvement was observed over time in terms of ROM in both groups.

### 3.5. Muscle Strength and Performance

No significant differences (*p* > 0.05) were observed between the two groups in terms of five-repetition sit-to-stand and isometric muscle strength on the symptomatic and asymptomatic side, neither in knee flexion nor in knee extension at baseline. However, a significant interaction effect (*p* < 0.001) was found between time and group for isometric muscle strength on the symptomatic as well as asymptomatic side, in knee flexion and extension both, as shown in Table 3. On the contrary, for five-repetition sit-to-stand, even though no significant interaction effect was found between time and group [F(1,38) = 0.07, *p* = 0.793, η^2^p = 0.002], a significant main effect for time [F(1,38) = 699.069, *p* = <0.001, η^2^p = 0.948] and treatment group [F(1,38) = 8.432, *p* = 0.006, η^2^p = 0.182] was reported.

### 3.6. Body Composition

No significant differences (*p* > 0.05) were observed between the two groups in terms of phase angle, impedance, visceral fat, percentage body fat, ICW ratio, skeletal muscle mass, segmental lean mass, and segmental fat mass at baseline. However, a statistically significant interaction effect (*p* < 0.05) was found between time and group for all body composition outcome variables as shown in Table 4, indicating that body composition parameters differed over time between the control and experimental (creatine) groups, and even though both groups showed a significant improvement in body composition, the creatine group demonstrated greater improvement.

## 4. Discussion

The purpose of this study was to determine the additive benefits of creatine supplementation when administered in combination with conventional physical therapy and resistance exercise training in patients with KOA. Statistically significant improvements were observed in both groups in terms of VAS and KOOS scores after four weeks of treatment, and the creatine group showed significantly greater improvements in terms of the VAS and KOOS scores as compared to the control group. However, no significant differences were observed in the KOOS-QOL subscale between the two groups after four weeks of treatment. Even though the creatine group showed a greater reduction in pain (VAS) compared to the control group, with a mean difference of 0.8 cm between the two groups, this difference did not meet the minimal clinically important difference (MCID) of 2.26 cm reported in the literature, suggesting that while there was greater improvement in the creatine group, it may not be large enough to be clinically meaningful [44]. Similarly, in the KOOS subscales, the creatine group demonstrated greater improvements than the control group, with mean differences of 12.55 for symptoms, 7.85 for pain, 7.80 for ADL, 9.75 for sports and recreation, and 3.35 for QOL. When compared to the minimal important difference (MID) values of 12.5 for symptoms, 11.8 for pain, 2.5 for ADL, 17.5 for sports and recreation, and 6.5 for QOL [45], the improvements in symptoms and ADL exceeded the MID, indicating clinically meaningful benefits. However, the improvements in pain, sports and recreation, and QOL did not reach the MID threshold, suggesting that while creatine supplementation had a positive effect, it may not be substantial enough for patients to perceive a noticeable benefit over the standard treatment in terms of the aforementioned outcomes. These findings indicate that creatine supplementation may provide additional benefits for symptom relief and functional improvement in daily activities for individuals with KOA, but its effects on pain, sports-related function, and overall quality of life are either clinically not significant or may require a longer treatment duration to achieve clinically meaningful improvements. Similarly, a study conducted by M Neves Jr et al. in 2011 [46] determined the effects of lower extremity resistance training with and without creatine supplementation on postmenopausal women with KOA and found the creatine supplementation group to be effective in improving WOMAC physical function, stiffness, and quality of life subscale scores. A significant improvement was observed in terms of pain in both groups, without any significant difference between the two groups after 12 weeks [46]. Another study conducted by Peeler J et al. compared the effects of creatine supplementation with lower body positive pressure treadmill training and exercise only group, but unlike the current study, positive effects of creatine supplementation were observed only in terms of the KOOS-QOL subscale after 12 weeks of treatment. An improvement in pain (VAS) was also observed in the creatine group but was not confirmed by longitudinal analysis; however, it was suggested that a larger sample may support the beneficial effects of creatine supplementation, as the sample group size was very small (*n* = 7), signifying a higher chance of false negatives. It is also important to point out that the participants were allowed to take NSAIDs during the study [43]. Cornish SM and Peeler JD also determined the beneficial effects of creatine supplementation, in comparison to control and found no significant improvement in terms of the KOOS scores; however, it is imperative to point out that participants did not perform any exercise in combination with creatine supplementation in that study [47].

In terms of range of motion, both the treatment groups showed statistically significant improvements; however, no significant differences were observed between the two groups, signifying that the improvement in knee range of motion was due to joint mobilization and exercise and had no additive benefit from creatine supplementation. Likewise, the existing literature has also shown manual therapy and exercise to be effective in pain, function, disability, and performance in persons with KOA [10,48]. Furthermore, in terms of isometric muscle strength and the five-repetition sit-to-stand test, statistically significant improvements were observed for both groups after four weeks of treatment, with the creatine supplementation group being superior to the control group at both follow-up intervals. These findings were in accordance with the findings of M Neves JR et al., showing the creatine supplementation group to be superior to the exercise only group in terms of the timed stand test (*p* = 0.004) [46]. Furthermore, both groups showed a statistically significant improvement in terms of 1-repetition maximum during the leg press exercise; however, no significant difference was observed between the two groups [35]. Moreover, according to the findings of Cornish SM and Peeler JD, in the absence of resistance exercise training, no positive effects of creatine supplementation were observed in terms of isometric muscle strength at 0°, 45°, and 90° of knee joint flexion and extension, even after 12 weeks [47]. Thus, it can be suggested that creatine supplementation augments the benefits of resistance exercise training in terms of muscle strength and function but may not be effective in the absence of the latter.

In terms of body composition analysis, statistically significant improvements were observed in both groups after four weeks of treatment, in terms of phase angle, impedance, visceral fat, percentage body fat, ICW ratio, skeletal muscle mass, segmental lean mass, and segmental fat mass. Moreover, a statistically significant difference was observed for all body composition outcomes between the two treatment groups in favor of the creatine supplementation group. The findings of the current study are similar to that of M Neves Jr et al., who also showed additive benefits of creatine supplementation in addition to lower extremity resistance exercise training in terms of lower extremity lean mass [46]. Moreover, the literature also suggests creatine supplementation to increase fat-free mass [49]. Furthermore, studies have shown that creatine acts as an osmolyte and increase water uptake in cells, which has been observed to augment protein synthesis and increase muscle mass and strength [50,51], similar to the findings of the current study showing higher overall and segmental ICW ratios in the creatine group. Furthermore, an increase in ICW ratio is directly proportional to a decrease in ECW ratio, the latter of which is associated with muscle atrophy and sarcopenia [52], which further advocates the beneficial effect of CS in addition to resistance exercise training. Even though creatine supplementation has been observed to have anti-inflammatory properties [53], previous studies conducted on persons with KOA have shown no positive effects on inflammation or stress markers [43]. It is also imperative to point out that the duration of treatment and follow-up of four weeks in the current study was very short in terms of truly analyzing the effects on muscle size, as it may take 8–12 weeks to undergo structural changes in muscles, and the initial increase in muscle strength has a major contribution from neural factors and increased neural drive rather than increase in muscle size [54]. Similarly, the percentage increase in lower extremity lean mass can also be partially attributed to increase in ICW ratio. This relatively short duration of the intervention (four weeks) may be viewed as a limitation, as longer trials (8–12 weeks) are generally required to observe substantial hypertrophic or long-term functional changes in muscles. However, this study aimed to examine early responses and feasibility in a low-resource context where lack of funding, participant adherence, and clinical expectations favor shorter interventions. Unfortunately, conducting a longer follow-up was not feasible due to resource constraints and limited participant compliance, as the research was carried out in a low-resource setting (Pakistan) and was mostly self-funded. Moreover, patients in this context often seek short-term symptomatic relief to quickly return to work and daily responsibilities, making prolonged follow-up challenging. Nonetheless, we agree that assessing the long-term durability and retention of benefits is crucial. This study therefore provides preliminary data and serves as a foundation for future research to evaluate sustained outcomes through extended follow-up durations.

Although creatine supplementation may increase the overall cost of KOA management, its ability to enhance the effects of resistance training, which is a key component of KOA treatment suggests that it could possibly be a cost-effective approach. Creatine plays this critical role in augmenting the effects of resistance training as it exists in muscles in the form of phosphocreatine (PCr), which serves as a rapidly accessible energy source during high-intensity, short-duration activities such as resistance training [12,25,26]. By increasing PCr availability, creatine supplementation augments ATP regeneration, allowing individuals to withstand greater force production, perform higher number of repetitions, and increase overall training volume [12,25,26]. This enhanced exercise capacity leads to greater strength gains and muscle hypertrophy over time, and patients may require a lesser number of treatment sessions to achieve their treatment goals [12,25,26]. However, this potential benefit warrants further studies focusing on long-term follow-up and the cost-effectiveness of creatine supplementation.

Regarding the potential adverse effects of creatine supplementation, research indicates that it may lead to weight gain; however, this increase is attributed to muscle mass rather than fat accumulation [12,25]. Additionally, creatine supplementation has been found to enhance water retention within muscles, leading to an ICW, which may also contribute to a rise in the overall body mass [12,26]. This rise in ICW has led to the assumption that creatine supplementation causes water retention. However, creatine itself is an osmotically active compound, resulting in a short-term increase in ICW rather than generalized water retention [12,26]. This acts as a stimulus for muscle growth and is linked to increased strength and improved functional capacity [51]. Over the long term, increase in total body water relative to muscle mass does not occur, indicating that creatine supplementation may not lead to water retention [12,26]. In fact, a meta-analysis by Forbes SC et al. (2019) demonstrated that creatine supplementation, when combined with resistance training, contributes to a reduction in fat mass in older adults [55]. Anecdotal reports suggest that creatine supplementation may cause side effects such as kidney damage; however, no scientific evidence has been found to support these claims [12,25,26]. Additionally, studies have shown no significant effects of creatine supplementation on renal or hepatic toxicity [46,56,57]. A meta-analysis published in 2019 found that CS does not cause renal damage, as it does not lead to significant changes in serum creatinine or plasma urea levels [56]. Moreover, in the context of KOA, Neves et al. reported no significant difference in creatinine clearance between the creatine supplementation group and the placebo group [46]. Thus, based on the available literature, creatine supplementation can be considered safe, with no significant evidence of renal or hepatic toxicity.

Based on the findings of the present study and the existing literature, creatine supplementation appears to be safe and demonstrates positive effects when combined with resistance exercise training. However, in the absence of resistance training, creatine may not provide additional benefits. Therefore, incorporating creatine supplementation alongside physical therapy and resistance exercise training may be a valuable approach in the management of individuals with KOA. Nonetheless, high-quality clinical trials with longer treatment durations and extended follow-up periods are required to further investigate its cost-effectiveness, long-term efficacy, and clinical relevance. Lastly, in the current study, hand-held modified sphygmomanometer test and bio-electrical impedance analysis were used for quantifying muscle strength and body composition parameters, and it is suggested that future studies should incorporate more specialized tools such as isokinetic dynamometer, electromyography, and magnetic resonance imaging (MRI) for quantifying muscle strength, performance, and body composition.

## 5. Conclusions

Physical therapy and resistance exercise training play a crucial role in the management of KOA by improving pain, function, range of motion, muscle strength, and body composition. However, creatine supplementation has the potential to offer additive benefits, particularly in improving muscle strength, skeletal muscle mass, intracellular water ratio, and functional performance. These findings suggest that creatine supplementation can act as a safe and potentially valuable adjunct to conventional physical therapy, primarily when combined with resistance exercise. However, in the absence of exercise, it may not be effective. Longer duration studies with larger samples are needed to confirm clinical relevance, assess long-term effects, and evaluate cost-effectiveness. Future studies should also consider advanced assessment tools, such as isokinetic dynamometry, electromyography, and MRI, to more precisely quantify muscle strength, function, and body composition.

## Figures and Tables

**Figure 1 jcm-14-08538-f001:**
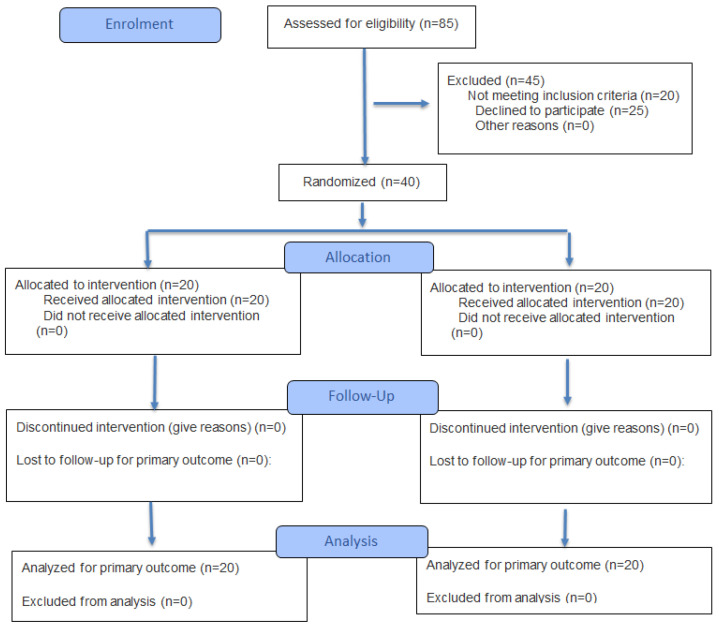
CONSORT diagram representing the flow of participants through the course of the study.

**Figure 2 jcm-14-08538-f002:**
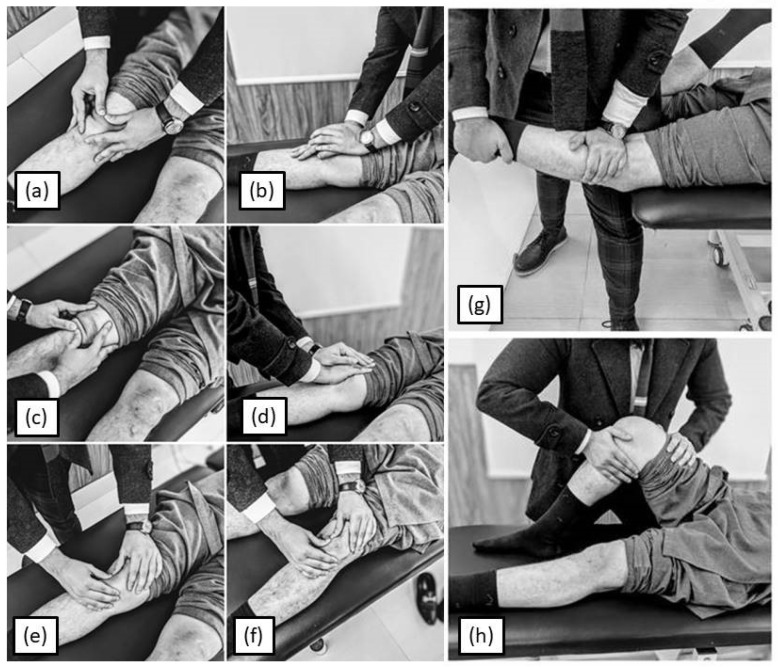
Manual therapy techniques administered at the knee–joint complex—(**a**,**b**) patello-femoral inferior glide to improve flexion, (**c**,**d**) patello-femoral superior glide to improve extension, (**e**) patello-femoral medial glide and (**f**) patello-femoral lateral glide to correct patellar mal-tracking, (**g**) tibio-femoral anterior glide to improve knee extension, and (**h**) tibio-femoral posterior glide to improve flexion.

**Figure 3 jcm-14-08538-f003:**
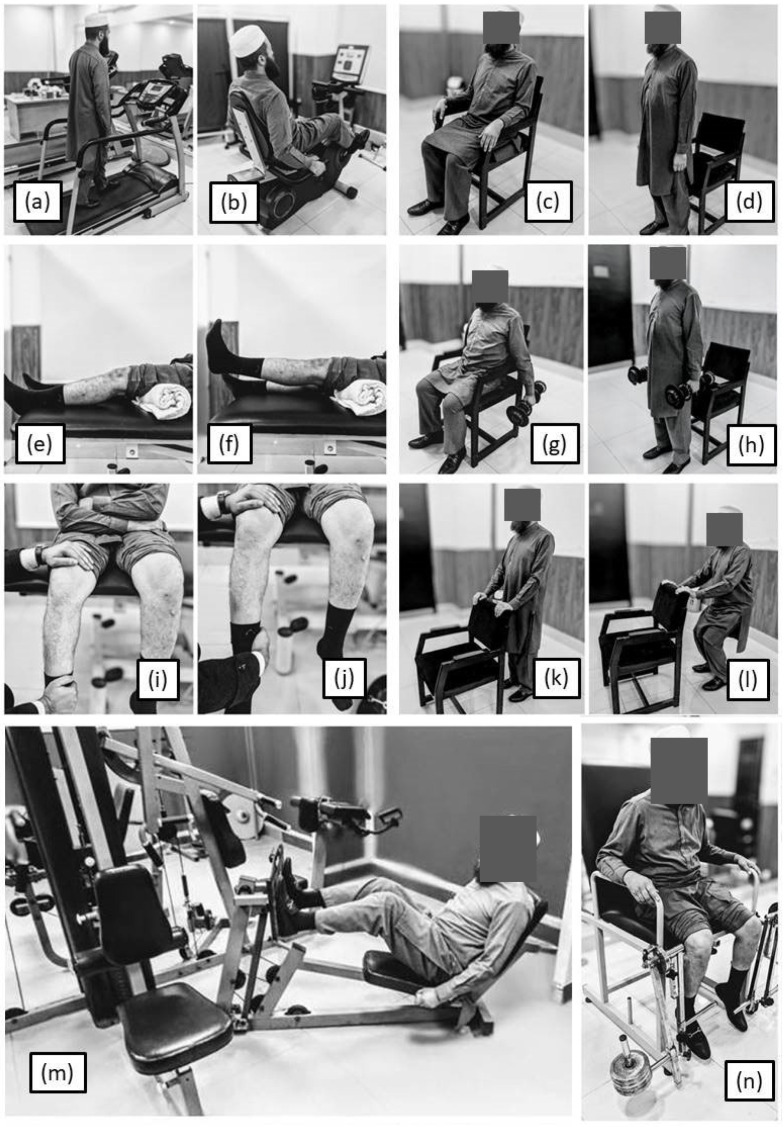
Exercise Protocol—(**a**) self-paced walking on treadmill, (**b**) cycling against maximal resistance, (**c**,**d**) sit-to-stand, (**e**) isometric knee extension, (**f**) terminal knee extension in lying, (**g**,**h**) sit-to-stand with weight, (**i**) isometric knee extension in sitting against manual resistance, (**j**) isometric knee flexion in sitting against manual resistance, (**k**,**l**) mini squats, (**m**) leg press exercise, and (**n**) knee extension exercise against resistance.

**Figure 4 jcm-14-08538-f004:**
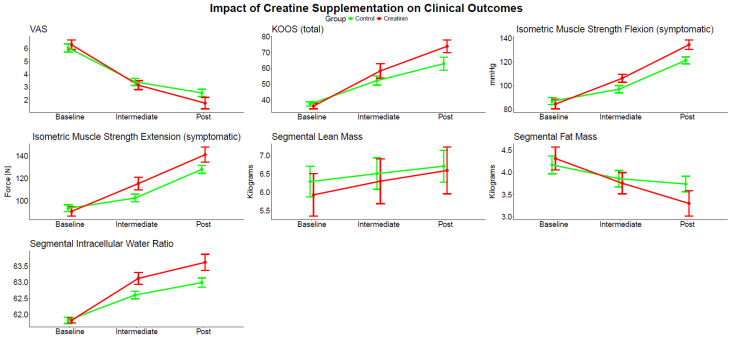
Effects on clinical outcomes in experimental (creatine) and control group after four weeks of treatment.

**Table 1 jcm-14-08538-t001:** Distribution of participants based on grade of knee osteoarthritis (KOA) and gender.

Variable	Control Group						Experimental (Creatine) Group						*p*-Value
Grade of KOA	I		II		III		I		II		III		0.60
	3		17		0		3		16		1		
Gender	Male			Female			Male			Female			1.00
	5			15			5			15			

**Table 2 jcm-14-08538-t002:** Individual characteristics of participants.

Variable	Control Group (Mean ± S.D)	Experimental Group (Mean ± S.D)	Mean Difference	*p*-Value
Age (years)	57.65 ± 5.65	58.1 ± 7.67	−0.45 [−4.77; 3.87]	0.83
Weight (kg)	75.20 ± 6.95	75.13 ± 8.55	0.08 [−4.91; 5.06]	0.98
Height (cm)	163.45 ± 8.87	164.95 ± 6.84	−1.50 [−6.57; 3.57]	0.55
BMI (kg/m^2^)	28.32 ± 3.69	27.64 ± 3.06	0.68 [−1.49; 2.85]	0.53

S.D: Standard deviation, BMI: body mass index.

**Table 3 jcm-14-08538-t003:** Comparison of VAS, fall risk score, ROM, IMS, 5xSST, and KOOS between the control and experimental (creatine) groups.

Variable		Control Group				Experimental (Creatine) Group				Interaction Effect (Time × Group)		
		Baseline (Mean ± S.D)	Follow-Up (Mean ± S.D)	Mean Difference	*p*-Value	Baseline (Mean ± S.D)	Follow-Up (Mean ± S.D)	Mean Difference	*p*-Value	F(1,38)	*p*-Value	Effect Size (η^2^_p_)
VAS (cm)		6.05 ± 0.69	2.55 ± 0.60	3.50 [3.18; 3.82]	<0.001	6.30 ± 0.80	1.75 ± 0.97	4.55 [4.04; 5.07]	<0.001	13.113	0.001	0.257
Fall Risk Score		3.34 ± 0.46	3.15 ± 0.44	0.19 [0.12; 0.25]	<0.001	3.41 ± 4.16	2.33 ± 0.89	1.08 [0.73; 1.42]	<0.001	28.381	<0.001	0.428
Five-repetition sit-to-stand test (seconds)		50.37 ± 5.28	18.43 ± 1.33	31.94 [28.93; 34.94]	<0.001	47.38 ± 7.52	16.07 ± 2.26	31.31 [27.30; 35.31]	<0.001	0.070	0.793	0.002
ROM (°)	Flexion Symptomatic Side	97.40 ± 7.84	137.40 ± 6.49	−40.00 [−41.31; −38.68]	<0.001	95.30 ± 12.17	135.00 ± 8.60	−39.70 [−43.44; −35.96]	<0.001	0.025	0.875	0.001
	Flexion Asymptomatic Side	130.80 ± 5.43	139.20 ± 5.30	−8.40 [−9.71; −7.08]	<0.001	127.30 ± 7.44	138.75 ± 6.25	−11.45 [−14.07; −8.83]	<0.001	4.727	0.036	0.111
	Extension Symptomatic Side	−10.25 ± 3.02	−1.10 ± 2.02	−9.15 [−10.13; −8.17]	<0.001	−10.50 ± 4.56	−2.00 ± 2.99	−8.5 [−9.60; −7.40]	<0.001	0.856	0.361	0.022
	Extension Asymptomatic Side	−3.05 ± 1.36	−0.45 ± 0.83	−2.60 [−3.04; −2.12]	<0.001	−2.80 ± 2.07	−0.70 ± 0.98	−2.1 [−2.76; −1.44]	<0.001	1.740	0.195	0.044
IMS (mmHg)	Flexion Symptomatic Side	87.05 ± 6.23	121.30 ± 6.17	−34.25 [−35.12; −33.38]	<0.001	84.60 ± 8.74	134.50 ± 8.57	−49.90 [−53.94; −45.86]	<0.001	62.905	<0.001	0.623
	Flexion Asymptomatic Side	104.75 ± 5.95	143.10 ± 6.73	−38.35 [−39.92; −36.78]	<0.001	102.05 ± 7.78	150.00 ± 9.73	−47.95 [−50.89; −45.01]	<0.001	36.423	<0.001	0.489
	Extension Symptomatic Side	93.50 ± 6.51	128.15 ± 7.43	−34.65 [−36.64; −33.66]	<0.001	90.75 ± 9.07	141.25 ± 14.41	−50.50 [−56.66; −44.34]	<0.001	28.248	<0.001	0.426
	Extension Asymptomatic Side	114.30 ± 7.61	159.00 ± 9.03	−44.70 [−45.79; −43.61]	<0.001	111.50 ± 13.87	172.25 ± 15.68	−60.75 [−66.39; −55.21]	<0.001	35.391	<0.001	0.482
KOOS	Total	37.40 ± 2.98	16.07 ± 2.26	−25.45 [−29.38; −21.52]	<0.001	36.10 ± 3.86	73.85 ± 8.62	−37.75 [−41.98; 33.52]	<0.001	19.874	<0.001	0.343
	Symptoms	48.15 ± 6.32	81.05 ± 9.05	−32.90 9 [−37.31; −28.49]	<0.001	45.25 ± 8.47	93.60 ± 8.47	−48.35 [−54.16; −42.54]	<0.001	19.641	<0.001	0.341
	Pain	42.70 ± 1.87	73.40 ± 10.54	−30.70 [−35.42; −25.98]	<0.001	42.80 ± 1.88	81.25 ± 7.45	−38.45 [−41.66; −35.24]	<0.001	8.084	0.007	0.175
	ADL	50.90 ± 1.29	75.05 ± 8.54	−24.15 [−28.09; −20.21]	<0.001	50.50 ± 1.88	82.85 ± 7.04	−32.35 [−35.8-; −28.90]	<0.001	10.758	0.002	0.221
	Sports and Recreation	9.50 ± 2.24	28.75 ± 10.75	−19.25 [−24.14; −14.09]	<0.001	9.00 ± 3.08	38.50 ± 13.29	−29.50 [−36.20; −22.80]	<0.001	6.430	0.015	0.145
	QOL	36.55 ± 5.73	69.30 ± 1.34	−32.75 [−35.31; −30.19]	<0.001	34.85 ± 6.42	72.65 ± 6.40	−37.80 [−42.37; −33.23]	<0.001	4.077	0.051	0.097

S.D: Standard deviation, VAS: visual analog scale, ROM: range of motion, IMS: isometric muscle strength, KOOS: Knee Osteoarthritis and Outcome Score, ADL: activities of daily living, QOL: quality of life. *p* < 0.05 is considered significant.

**Table 4 jcm-14-08538-t004:** Comparison of body composition parameters between the control and experimental (creatine) groups.

Variable	Control Group				Experimental (Creatine) Group				Interaction Effect (Time × Group)		
	Baseline (Mean ± S.D)	Follow-Up (Mean ± S.D)	Mean Difference	*p*-Value	Baseline (Mean ± S.D)	Follow-Up (Mean ± S.D)	Mean Difference	*p*-Value			
									F(1,38)	*p*-Value	Effect Size (η^2^_p_)
Phase Angle (°)	5.38 ± 0.47	5.68 ± 0.47	−0.30 [−0.30; −0.30]	<0.001	5.28 ± 0.40	5.89 ± 0.48	−0.61 [−0.73; −0.49]	<0.001	29.315	<0.001	0.435
Impedance (Rt. Leg)	280.37 ± 51.54	272.54 ± 51.93	7.83 [7.12; 8.53]	<0.001	295.76 ± 56.88	283.91 ± 57.07	11.85 [10.12; 13.59]	<0.001	20.20	<0.001	0.347
Impedance (Lt. Leg)	286.14 ± 44.50	283.81 ± 44.55	2.34 [2.26; 2.41]	<0.001	301.33 ± 50.46	296.69 ± 50.48	4.64 [3.81; 5.46]	<0.001	33.894	<0.001	0.471
Visceral Fat	148.76 ± 14.86	143.72 ± 15.07	5.04 [3.91; 6.16]	<0.001	154.76 ± 19.74	144.28 ± 19.31	10.48 [8.03; 12.93]	<0.001	17.826	<0.001	0.319
Percentage Body Fat (%)	37.17 ± 6.33	36.30 ± 6.37	0.87 [0.69; 1.05]	<0.001	38.97 ± 7.04	35.98 ± 6.94	2.99 [2.00; 3.98]	<0.001	19.482	<0.001	0.339
ICW Ratio (%)	61.83 ± 0.21	62.99 ± 0.30	−1.16 [−1.29; −1.03]	<0.001	61.83 ± 0.17	63.62 ± 0.53	−1.79 [−2.10; −1.48]	<0.001	15.195	<0.001	0.286
ICW Ratio (Rt. Leg) (%)	61.93 ± 0.23	63.12 ± 0.31	−1.20 [−1.32; −1.07]	<0.001	61.92 ± 0.16	63.71 ± 0.53	−1.79 [−2.10; −1.48]	<0.001	1.761	0.001	0.266
ICW Ratio (Lt. Leg) (%)	61.63 ± 0.22	62.83 ± 0.31	−1.20 [−1.32; −1.07]	<0.001	61.63 ± 0.16	63.42 ± 0.54	−1.79 [−2.11; −1.48]	<0.001	13.861	0.001	0.267
Skeletal Muscle Mass (kg)	19.22 ± 4.32	19.69 ± 4.28	−0.46 [−0.60; −0.33]	<0.001	18.30 ± 3.94	22.53 ± 4.21	−4.23 [−5.34; −3.11]	<0.001	49.262	<0.001	0.565
Segmental Lean (Rt. Leg)	6.57 ± 1.03	7.00 ± 1.09	−0.43 [−0.46; −0.40]	<0.001	6.20 ± 1.38	6.89 ± 1.51	−0.69 [−0.80; −0.58]	<0.001	22.099	<0.001	0.368
Segmental Lean (Lt. Leg)	6.01 ± 0.96	6.42 ± 0.98	−0.41 [−0.43; −0.39]	<0.001	5.66 ± 1.24	6.30 ± 1.36	−0.64 [−0.74; −0.54]	<0.001	22.047	<0.001	0.367
Segmental Fat (Rt. Leg)	4.16 ± 0.42	3.73 ± 0.37	0.43 [0.39; 0.46]	<0.001	4.30 ± 0.55	3.30 ± 0.61	1.01 [0.77; 1.24]	<0.001	26.412	<0.001	0.410
Segmental Fat (Lt. Leg)	4.17 ± 0.44	3.74 ± 0.38	0.43 [0.39; 0.47]	<0.001	4.32 ± 0.55	3.30 ± 0.61	1.02 [0.79; 1.25]	<0.001	27.521	<0.001	0.420

S.D: Standard deviation, Rt.: right, Lt.: left. *p* < 0.05 is considered significant.

## Data Availability

The data presented in this study are available on request from the corresponding author.

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
