# Peer review of "Additional Benefits of Creatine Supplementation with Physical Therapy and Resistance Exercise in Knee Osteoarthritis: A Randomized Controlled Trial"

_jcm, 2025, doi:10.3390/jcm14238538_

Round 1

Reviewer 1 Report

Comments and Suggestions for Authors

This study aimed to evaluate the additive benefits of creatine supplementation, when combined with resistance exercise training and physical therapy, in the management of knee OA.

This is an important and interesting topic, as it refers to treatment for knee OA, which is a very common musculoskeletal disorder.  The study design is well-structured, featuring two independent groups: a study group receiving Creatine and a placebo control group. Both groups received similar additional PT treatments.

I think that some issues should be further clarified before publication, and  I have some specific comments:  

  1. The aim should be written in the abstract, separate from the background
  2. Abstract – should add that it was a placebo control group, and that PT included.. (electro, manual... etc).
  3. The rationale for adding creatine is not clear enough. What is creatine used for? What are its effects? It should be explained in the introduction.
  4. Methods – “knee pain no more than 8/10”.. did you also have a minimum limit for pain to be included?
  5. Methods – “grade III or less on Kellgren” – who made the diagnosis?
  6. Methods – “The active comparator group received a placebo (maltodextrin packets) to ensure blinding.” – this is not mentioned in the abstract or earlier. maltodextrin packets – clarify why this was used for the placebo group.
  7. Fig 1 should include only the manual technique. And Figure 2 should include the exercises. In Figure 1 you mixed manual and exercises, which confuses the purpose of the figure.
  8. Methods - sphygmomanometer test is not clear enough. Please describe and clarify the test, what you measured and in what units.

In the table you mention IMS and mmHg – in the text you don’t describe this at all. Clarify how you measured flexion and extension muscle strength; you mention it was isometric – against what resistance?

  1. Did you monitor for the creatine consumption? The participants needed to record in any application /diary?
  2. Fall of risk – how did you measure?, this is not described in the methods.
  3. KOOS – should be clarified, what it measures, and what is the scale (0-to?) – higher score represents..?
  4. Home exercises consisted .... isometric (against manual resistance_ - how did the participants perform exercise against resistance at home?
  5. Table 3 – bold or mark the significance p-value.
  6. Table 3+ 4 – any abbreviation should be explained at the bottom of the table (even if mentioned in the text itself), the significance level should also be written.
  7. Conclusion – study limitations should be added at the end of the conclusion. Although it is hidden in terms of future studies, this should be addressed.
  8. Figure – I think the figure quality should be improved.

Author Response

We would like to sincerely thank the reviewers for their time, effort, and constructive comments on our manuscript. We greatly appreciate the detailed feedback, which has helped us clarify and improve various aspects of the study. The detailed comment by comment responses to the respected reviewers’ comments are as follows:

Reviewers Comment: The aim should be written in the abstract, separate from the background

Response: Aims/Objectives: To determine the additive effects of creatine supplementation alongside physical therapy (PT) and resistance exercise training in individuals with knee osteoarthritis (Page 1, Line 19–20).

Reviewer Comment: Abstract – should add that it was a placebo control group, and that PT included.. (electro, manual... etc).

Response: Participants were randomly allocated to either a placebo control group, which received placebo supplementation (maltodextrin) along with PT, including heat therapy, electrotherapy, manual therapy, and resistance exercises, for four weeks, or an experimental group, which received CS instead of maltodextrin in addition to the same treatment. (Page 1, Line 22–26).

Reviewer Comment: The rationale for adding creatine is not clear enough. What is creatine used for? What are its effects? It should be explained in the introduction.

Response: Effects of exercise training can be augmented by special diet or external supple-mentation, and previous research has shown that creatine is one of the most common and effective supplement in sports and rehabilitation to augment resistance training for improving muscular strength and performance [13, 26]. Creatine is synthesized from amino acids, and is stored in skeletal muscles as free creatine and phosphocreatine (PCr) [13, 27, 28]. Within muscle tissue, creatine supports rapid energy regeneration, as PCr serves as an immediate energy reserve, allowing quick ATP re-synthesis during short, high intensity activities such as resistance exercise and strength training. By increasing PCr availability, CS enhances ATP production, enabling greater force generation and increased exercise volume, capacity and intensity. Over time, this con-tributes to enhanced muscle strength and hypertrophy [13, 27, 28]. Furthermore, CS supports muscle repair and growth by stimulating satellite cell signaling and increasing intracellular water (ICW), leading to muscle cell expansion [13, 27, 28].. It may also reduce muscle breakdown by lowering myostatin levels, thus supporting lean mass development. Moreover, studies show that CS facilitates recovery, prevents injury, and displays neuro-protective and anti-inflammatory effects [13, 27, 28]. The potential therapeutic benefits of CS have been reported in several conditions such as muscular dystrophy, Parkinson’s disease, stroke, and even in aging and pregnancy [13, 27, 28]. (Page 2, Line 73–87).

Reviewer Comment: Methods – “knee pain no more than 8/10”.. did you also have a minimum limit for pain to be included?

Response: knee pain no less than 4/10 and no greater than 8/10 on visual numeric pain rating scale (VNRS) during the last week (Page 4, Line 123–125).

Reviewer Comment: Methods – “grade III or less on Kellgren” – who made the diagnosis?

Response: Patients were referred by a physiatrist and had a confirmed diagnosis of KOA based on radiographic findings. The X-ray reports were issued and graded by a radiologist, and included the severity grade of KOA. (Page 4, Line 126–128).

Reviewer Comment: Methods – “The active comparator group received a placebo (maltodextrin packets) to ensure blinding.” – this is not mentioned in the abstract or earlier. maltodextrin packets – clarify why this was used for the placebo group.

Response: The control group received a placebo (maltodextrin packets) in the same pattern to ensure blinding. Maltodextrin was used as the placebo as it closely matches the appearance, texture, and solubility of creatine monohydrate, ensuring effective blinding without producing any physiological effects at the administered dose. Moreover, several previous studies investigating CS have also employed maltodextrin as a placebo for the control groups [13]. Participants were provided with pre-weighed, individually labeled sachets of creatine monohydrate (or placebo) for daily consumption, and were instructed to record their intake in a supplementation and exercise log diary, which was reviewed weekly by the research team during supervised physical therapy sessions. (Page 4–5, Line 167–176).

Reviewer Comment: Fig 1 should include only the manual technique. And Figure 2 should include the exercises. In Figure 1 you mixed manual and exercises, which confuses the purpose of the figure.

Response: The suggested changes have been mentioned in Figure 2 (Page 6) and Figure 3 (Page 8).

Reviewer Comment: Methods - sphygmomanometer test is not clear enough. Please describe and clarify the test, what you measured and in what units.

Response: Isometric muscle strength of the knee flexors and extensors was assessed using a modified sphygmomanometer test following procedures adapted from Silva BBC et al [32]. Participants were instructed to sit upright with hips and knees flexed to 90°, feet unsupported, and arms crossed. A stabilization belt was applied around the distal third of the leg and bed. For knee flexion, the sphygmomanometer cuff was placed on the posterior aspect of the distal part of the leg against the bed, and for extension, on the anterior aspect of the distal part of the leg against the belt. After a familiarization trial, participants performed one maximal 5-second isometric contraction, cued verbally, and instructed to avoid the Valsalva maneuver. The peak pressure (mmHg) was recorded as the isometric muscle strength value. The procedure is shown to have high inter-rater and intra-rater reliability for both knee flexors and extensors with an ICC > 0.83 [32]. (Page 6, Line 201–212)

Reviewer comment: In the table you mention IMS and mmHg – in the text you don’t describe this at all. Clarify how you measured flexion and extension muscle strength; you mention it was isometric – against what resistance?
Response: Isometric muscle strength of the knee flexors and extensors was assessed using a modified sphygmomanometer test following procedures adapted from Silva BBC et al [32]. Participants were instructed to sit upright with hips and knees flexed to 90°, feet unsupported, and arms crossed. A stabilization belt was applied around the distal third of the leg and bed. For knee flexion, the sphygmomanometer cuff was placed on the posterior aspect of the distal part of the leg against the bed, and for extension, on the anterior aspect of the distal part of the leg against the belt. After a familiarization trial, participants performed one maximal 5-second isometric contraction, cued verbally, and instructed to avoid the Valsalva maneuver. The peak pressure (mmHg) was recorded as the isometric muscle strength value. The procedure is shown to have high inter-rater and intra-rater reliability for both knee flexors and extensors with an ICC > 0.83 [32]. (Page 6, Line 201–212)

Reviewer Comment: Did you monitor for the creatine consumption? The participants needed to record in any application /diary?

Response: Participants were provided with pre-weighed, individually labeled sachets of creatine monohydrate (or placebo) for daily consumption, and were instructed to record their intake in a supplementation and exercise log diary, which was reviewed weekly by the research team during supervised physical therapy sessions. (Page 5, Line 172–176)

Reviewer Comment: Fall of risk – how did you measure?, this is not described in the methods.

Response: Fall risk was assessed using the Biodex Balance System (Biodex Medical Systems Inc., Shirley, NY, USA), which is shown to have a test–retest reliability ICC of 0.64–0.91 [39] and has been used previously in musculoskeletal conditions including KOA to evaluate balance impairments and fall risk [40-42]. For the fall risk assessment, par-ticipants were instructed to stand with their eyes open on an unstable platform set at stability level 6. Each participant completed three 20 second trials, the average of which was computed afterwards. Before the test, a familiarization trial was conducted to ensure participants understood the procedure, and those scores were not recorded. During the test, participants were instructed to maintain their balance. A higher score on the Biodex Balance System is indicative of a greater risk of falling [41, 43]. (Page 9, Line 246–255)

Reviewer Comment: KOOS – should be clarified, what it measures, and what is the scale (0-to?) – higher score represents..?

Response: KOOS, a self-administered questionnaire was used to quantify pain, symptoms, activities of daily living (ADL), sports and recreation function and quality of life (QOL), and has an ICC of 0.83-0.89 for KOA [31]. Each item was scored on a 5-point Likert scale, and the raw scores were transformed to a 0–100 scale, where a higher score signifies better outcome. (Page 5, Line 188–192)

Reviewer Comment: Home exercises consisted .... isometric (against manual resistance_ - how did the participants perform exercise against resistance at home?

Response: isometric (against manual resistance with the help of a family member) knee extension and flexion (Page 4, Line 157–158)

Reviewer Comment: Table 3 – bold or mark the significance p-value.

Response: The suggested changes have been made

Reviewer Comment: Table 3+ 4 – any abbreviation should be explained at the bottom of the table (even if mentioned in the text itself), the significance level should also be written.

Response: The suggested changes have been made

Reviewer Comment: Conclusion – study limitations should be added at the end of the conclusion. Although it is hidden in terms of future studies, this should be addressed.

Response: Physical therapy and resistance exercise training play a crucial role in the man-agement of KOA by improving pain, function, range of motion, muscle strength, and body composition. However, creatine supplementation has the potential to offer ad-ditive benefits, particularly in improving muscle strength, skeletal muscle mass, in-tracellular water ratio, and functional performance. These findings suggest that crea-tine supplementation can act as a safe and potentially valuable adjunct to conventional physical therapy, primarily when combined with resistance exercise. However, in the absence of exercise, its benefits may be limited. Longer duration studies with larger samples are needed to confirm clinical relevance, assess long term effects, and evaluate cost effectiveness. Future studies should also consider advanced assessment tools, such as isokinetic dynamometry, electromyography, and MRI, to more precisely quantify muscle strength, function, and body composition. (Page 19, Line 506–517)

Reviewer comment: Figure – I think the figure quality should be improved.

Response: Reviewer recommendations have been incorporated

Reviewer 2 Report

Comments and Suggestions for Authors

The study presents the effect of adding creatine supplementation to physiotherapy in patients with knee osteoarthritis (KOA). Given the high prevalence of KOA, the choice of research topic is justified and provides new insights into the potential benefits of supporting physiotherapy in the treatment of osteoarthritis.
Below, I provide my detailed comments regarding the manuscript.

Lines 16–19: The aim of the study should be clearly and explicitly defined in the Abstract.

Line 80: The reference to Figure 1 in the text is unclear. Please verify whether this is an error or an intentional placement.

Lines 89–90: Please specify the time frame for pain assessment — was it the pain on the day of examination, during the last week, month, etc.?

Lines 90–91: The inclusion of patients with Kellgren–Lawrence grade III and lower suggests that individuals with grade I may have been included. This may compromise the homogeneity of the study group.

Lines 104–122: Please clarify whether the patients performed the exercises while experiencing pain, and if so, what level of pain was considered acceptable during training.

Lines 303–304: The sentence appears to be misplaced on a new line. Please format the text correctly.

Author Response

We would like to sincerely thank the reviewers for their time, effort, and constructive comments on our manuscript. We greatly appreciate the detailed feedback, which has helped us clarify and improve various aspects of the study. The detailed comment by comment responses to the respected reviewers’ comments are as follows:

Reviewers Comment: Lines 16–19: The aim of the study should be clearly and explicitly defined in the Abstract.

Response: Aims/Objectives: To determine the additive effects of creatine supplementation alongside physical therapy (PT) and resistance exercise training in individuals with knee osteoarthritis (Page 1, Line 19–20).

Reviewer comment: Line 80: The reference to Figure 1 in the text is unclear. Please verify whether this is an error or an intentional placement.

Response: Figure 1 was supposed to be a CONSORT diagram. The aforementioned correction has been made.Thank you for pointing this out. This should have been CONSORT diagram

Reviewer Comment: Lines 89–90: Please specify the time frame for pain assessment — was it the pain on the day of examination, during the last week, month, etc.?

Response: knee pain no less than 4/10 and no greater than 8/10 on visual numeric pain rating scale (VNRS) during the last week (Page 4, Line 123–125)

Reviewer Comment: Lines 90–91: The inclusion of patients with Kellgren–Lawrence grade III and lower suggests that individuals with grade I may have been included. This may compromise the homogeneity of the study group.

Response: Yes but there was an equal distribution of Grade 1 KOA patients in both groups with no significant difference between the two group in terms of grade of KOA in the two groups as shown in Table 1, thus both groups were similar.

Reviewer Comment: Lines 104–122: Please clarify whether the patients performed the exercises while experiencing pain, and if so, what level of pain was considered acceptable during training.

Response: Treatment intensity was reduced in case of increase in pain or inability of patient to perform the exercise. If exercise still caused pain the specific exercise was halted and skipped for that session. (Page 4, Line 152–154)

Reviewer 3 Report

Comments and Suggestions for Authors

Dear Authors,

The topic is clinically relevant and timely — creatine supplementation in musculoskeletal rehabilitation is an underexplored but emerging area. However, the manuscript, while clearly written and methodologically structured, has significant methodological, analytical, and interpretative weaknesses that must be addressed.

Major issues

  • Sample size (n = 40) is very small for a randomized controlled trial assessing multiple outcomes with physiological and functional measures. The sample size calculation (based on VAS only) is not robust enough to justify generalization to several endpoints.

  • Randomization procedure is insufficiently described. There is no detail on sequence generation (e.g., computer randomization, permuted blocks) or allocation concealment.

  • Blinding: Although stated as “double-blind,” the trial design makes this questionable:

    • The creatine group had a clear loading phase (20 g/day in divided doses), which could produce noticeable gastrointestinal symptoms or water retention, easily unblinding participants and therapists.

    • The placebo (maltodextrin) is mentioned but not matched for taste, texture, or caloric content.

  • Duration (4 weeks) is too short to meaningfully evaluate muscle hypertrophy or sustained functional changes in osteoarthritis. Most prior creatine-exercise trials last 8–12 weeks minimum.

  • No follow-up period post-intervention — durability of the effects is unknown.

  • Heterogeneous interventions: Participants received both physical therapy and home exercise; adherence and fidelity of the home-based component were self-reported, introducing high bias potential.

  • Multiplicity of endpoints (VAS, KOOS subscales, fall risk, ROM, isometric strength, body composition) without predefined primary outcome inflates type I error risk.

  • Body composition via bioelectrical impedance (BIA) is insufficiently sensitive for short-term changes and confounded by hydration status, especially when using creatine (which alters intracellular water).

  • Isometric strength via sphygmomanometer is not a gold-standard measure; more reliable tools (isokinetic dynamometer) should be used or acknowledged as a limitation.

  • Fall Risk Score instrument not identified — unclear validity or clinical interpretation.

  • The analysis relies heavily on repeated t-tests and mixed ANOVA without correction for multiple outcomes or checking assumptions (sphericity, homogeneity).

  • The effect size reporting (η²p) is appropriate but not consistently interpreted (e.g., small/moderate/large).

  • Confidence intervals are sometimes misused — negative signs for differences are confusing; precision intervals overlap substantially, suggesting possible overinterpretation.

  • No intention-to-treat (ITT) analysis reported — necessary for RCT validity, even if no dropouts occurred.

  • The manuscript repeatedly claims “significant improvements” in multiple variables, but the magnitude of these improvements is statistically small and clinically negligible.

  • The discussion acknowledges the lack of minimal clinically important difference (MCID) in VAS, but other non-significant findings are still framed as positive — indicating interpretive bias.

  • The short duration (4 weeks) cannot support conclusions about muscle hypertrophy or body composition.

  • ROM improvements are likely attributable to therapy rather than creatine; yet the text implies causal contribution.

The authors cite literature (e.g., Neves et al., Peeler et al.) selectively, ignoring trials with null or conflicting results in older adults and OA populations.

English and style: Generally readable but too descriptive in sections of Results and Discussion; needs tighter scientific writing (avoid redundancy such as “significant improvements were observed…” repeated many times).

  • Figures and Tables: Contain excessive decimals and unnecessary precision (e.g., mean difference [−39.70; −35.96]).

  • Ethical statement: Appropriate, but the inclusion of “AI tools such as ChatGPT and Grammarly” should be checked against the journal’s disclosure policy (many high-impact journals require detailed AI use justification or disclaim it entirely).

  • References: Many self-citations (Osama et al.) and limited inclusion of high-quality RCTs or meta-analyses (e.g., no reference to latest 2023/2024 OA management guidelines or large creatine meta-analyses). Furthermore, you should cite the following articles: 10.3390/sports11040091; 10.3390/jpm14080784.

 Minor issues

  • Eligibility criteria lack clarity on baseline physical activity, supplement use, and comorbidities (e.g., diabetes, cardiovascular disease).

  • Ethical approval is appropriately reported but lacks a statement on data monitoring or adverse event reporting.

  • The inclusion of both mean differences and p-values without visual confidence intervals reduces interpretability.

  • Graphical presentation (Figures 3–4) lacks error bars and individual data points.

  • No adverse events or safety monitoring data reported, despite creatine’s known effects on renal parameters and water balance.

  • No registration summary from ClinicalTrials.gov included (should match NCT04423887 record).

  • CONSORT checklist and flow diagram (Figure 3) incomplete — no numbers for assessed, excluded, or analyzed participants.

  • No cost-benefit analysis or subgroup analyses (e.g., by gender, age, KOA grade).

The conclusion overstates the results. Based on the presented data:

  • Creatine may provide small additive short-term improvements in strength and pain perception when combined with resistance exercise, but the evidence is not robust or clinically meaningful.

  • The claim that creatine “offers significant additional benefits” should be moderated.

  • The manuscript should emphasize feasibility and pilot nature rather than efficacy.

Author Response

We would like to sincerely thank the reviewers for their time, effort, and constructive comments on our manuscript. We greatly appreciate the detailed feedback, which has helped us clarify and improve various aspects of the study. The detailed comment by comment responses to the respected reviewers’ comments are as follows:

Reviewer comment: Sample size (n = 40) is very small for a randomized controlled trial assessing multiple outcomes with physiological and functional measures. The sample size calculation (based on VAS only) is not robust enough to justify generalization to several endpoints.

Response: A sample size estimation was performed prior the experiments using the OpenEpi sample size calculator [44]. A total sample of 34 patients was calculated using the data from Peeler J et al [45] for VAS, with a mean 2.17±1.95 for control group and 0.7±0.85 for the experimental group, confidence interval of 95% and a power of 80%. A total of 40 participants (20 per group) were included in the study to compensate for any potential drop-out or loss in follow-up. The VAS was selected as the primary outcome measure because pain reduction was the principal clinical endpoint of interest and had the most robust prior data available for sample size estimation. Given the limited research on CS in KOA, and lack of reliable mean difference data for other outcomes were unavailable, additional outcomes were therefore exploratory, and the present study provides valuable preliminary data for future power calculations in this emerging research area. (Page 9, Line 257–268)

Reviewer Comment: Randomization procedure is insufficiently described. There is no detail on sequence generation (e.g., computer randomization, permuted blocks) or allocation concealment.

Response: Randomization was carried out using a simple randomization (coin toss) method by an independent researcher not involved in data collection or assessment. Group allocation (creatine supplementation or placebo) was concealed using opaque, sealed envelopes opened only after participant enrollment to ensure allocation concealment. Both participants and assessors were blinded to the treatment group allocation to minimize bias. (Page 3, Line 113–118)

Reviewer comment: Blinding: Although stated as “double-blind,” the trial design makes this questionable:

Response: Randomization was carried out using a simple randomization (coin toss) method by an independent researcher not involved in data collection or assessment. Group allocation (creatine supplementation or placebo) was concealed using opaque, sealed envelopes opened only after participant enrollment to ensure allocation concealment. Both participants and assessors were blinded to the treatment group allocation to minimize bias. (Page 3, Line 113–118)

The control group received a placebo (maltodextrin packets) in the same pattern to en-sure blinding. Maltodextrin was used as the placebo as it closely matches the appearance, texture, and solubility of creatine monohydrate, ensuring effective blinding without producing any physiological effects at the administered dose. Moreover, several previous studies investigating CS have also employed maltodextrin as a placebo for the control groups [13]. Participants were provided with pre-weighed, individually labeled sachets of creatine monohydrate (or placebo) for daily consumption, and were instructed to record their intake in a supplementation and exercise log diary, which was reviewed weekly by the research team during supervised physical therapy sessions.

Regarding tolerability, the loading dose (20 g/day divided into four 5 g doses) followed by maintenance dose (5 g/day) is consistent with standard guidelines of In-ternational Society of Sports Nutrition and have been shown to be well tolerated in with minimal gastrointestinal effects [13, 27, 28]. Furthermore, no participants reported adverse gastrointestinal symptoms or noticeable water retention in the current study, suggesting that unblinding was unlikely. (Page 4 and 5, Line 168–182)

Reviewer Comment: The creatine group had a clear loading phase (20 g/day in divided doses), which could produce noticeable gastrointestinal symptoms or water retention, easily unblinding participants and therapists.

Response: Regarding tolerability, the loading dose (20 g/day divided into four 5 g doses) followed by maintenance dose (5 g/day) is consistent with standard guidelines of In-ternational Society of Sports Nutrition and have been shown to be well tolerated in with minimal gastrointestinal effects [13, 27, 28]. Furthermore, no participants reported adverse gastrointestinal symptoms or noticeable water retention in the current study, suggesting that unblinding was unlikely. (Page 5, Line 177–182)

Reviewer Comment: The placebo (maltodextrin) is mentioned but not matched for taste, texture, or caloric content.

Response: The control group received a placebo (maltodextrin packets) in the same pattern to en-sure blinding. Maltodextrin was used as the placebo as it closely matches the appearance, texture, and solubility of creatine monohydrate, ensuring effective blinding without producing any physiological effects at the administered dose. Moreover, several previous studies investigating CS have also employed maltodextrin as a placebo for the control groups [13]. Participants were provided with pre-weighed, individually labeled sachets of creatine monohydrate (or placebo) for daily consumption, and were instructed to record their intake in a supplementation and exercise log diary, which was reviewed weekly by the research team during supervised physical therapy sessions.

Regarding tolerability, the loading dose (20 g/day divided into four 5 g doses) followed by maintenance dose (5 g/day) is consistent with standard guidelines of In-ternational Society of Sports Nutrition and have been shown to be well tolerated in with minimal gastrointestinal effects [13, 27, 28]. Furthermore, no participants reported adverse gastrointestinal symptoms or noticeable water retention in the current study, suggesting that unblinding was unlikely. (Page 4 and 5, Line 168–182)

Reviewer comment: Duration (4 weeks) is too short to meaningfully evaluate muscle hypertrophy or sustained functional changes in osteoarthritis. Most prior creatine-exercise trials last 8–12 weeks minimum.

Response: Yes and this is a major limitation of the study and has been acknowledged repeatedly in the manuscript:

It is also imperative to point out that the duration of treatment and follow up of 4 weeks in the current study was very short in terms of truly analyzing the effects on muscle size, as it may take 8 – 12 weeks to undergo structural changes in muscles, and the initial increase in muscle strength has a major contribution from neural factors and increased neural drive rather than increase in muscle size [56]. Similarly, the percentage increase in lower extremity lean mass can also be partially attributed to increase in ICW ratio. This relatively short duration of the intervention (four weeks) may be viewed as a limitation, as longer trials (8–12 weeks) are generally required to observe substantial hypertrophic or long-term functional changes in muscles. However, this study aimed to examine early responses and feasibility in a low-resource context where lack of funding, participant adherence, and clinical expectations favor shorter interventions. Unfortunately, conducting a longer follow-up was not feasible due to resource constraints and limited participant compliance, as the research was carried out in a low-resource setting (Pakistan) and was mostly self-funded. Moreover, patients in this context often seek short-term symptomatic relief to quickly return to work and daily responsibilities, making prolonged follow-up challenging. Nonetheless, we agree that assessing the long-term durability and retention of benefits is crucial. This study therefore provides preliminary data and serves as a foundation for future research to evaluate sustained outcomes through extended follow-up durations. (Page 17 and 18, Line 436–454)

Reviewer Comment: No follow-up period post-intervention — durability of the effects is unknown.

Response: Unfortunately, conducting a longer follow-up was not feasible due to resource constraints and limited participant compliance, as the research was carried out in a low-resource setting (Pakistan) and was mostly self-funded. Moreover, patients in this context often seek short-term symptomatic relief to quickly return to work and daily responsibilities, making prolonged follow-up challenging. Nonetheless, we agree that assessing the long-term durability and retention of benefits is crucial. This study therefore provides preliminary data and serves as a foundation for future research to evaluate sustained outcomes through extended follow-up durations. (Page 18, Line 447–454)

Reviewer Comment: Heterogeneous interventions: Participants received both physical therapy and home exercise; adherence and fidelity of the home-based component were self-reported, introducing high bias potential.

Response: We appreciate the reviewer’s observation. However, we would like to clarify that both groups received identical physical therapy protocols and home exercise regimens, with the only difference being the supplementation (creatine vs. placebo). Therefore, the intervention was homogeneous across groups, and any between-group differences can reasonably be attributed to the supplementation rather than variability in therapy.

To minimize bias, participants in both groups received the same instructions, exercise supervision during clinic sessions, and identical home exercise logs to record compliance. The therapist providing treatment was blinded to group allocation, and participants were instructed not to disclose their supplementation status. While self-reported adherence does have inherent limitations, it was uniform across both groups and thus unlikely to systematically bias the comparison.

Reviewer’s Comment: Multiplicity of endpoints (VAS, KOOS subscales, fall risk, ROM, isometric strength, body composition) without predefined primary outcome inflates type I error risk.

Response: We thank the reviewer for this insightful comment. However, we would like to clarify that pain intensity measured by the Visual Analogue Scale (VAS) was identified as the primary outcome variable for sample size calculation and hypothesis testing. The additional variables (KOOS subscales, fall risk, ROM, isometric strength, and body composition) were included as secondary outcomes to provide a broader understanding of the potential functional and physiological effects of creatine supplementation.

As shown in the manuscript on page:

“A sample size estimation was performed prior the experiments using the OpenEpi sample size calculator [45]. A total sample of 34 patients was calculated using the data from Peeler J et al [46] for VAS, with a mean 2.17±1.95 for control group and 0.7±0.85 for the experimental group, confidence interval of 95% and a power of 80%. A total of 40 participants (20 per group) were included in the study to compensate for any potential drop-out or loss in follow-up. The VAS was selected as the primary outcome measure because pain reduction was the principal clinical endpoint of interest and had the most robust prior data available for sample size estimation. Given the limited research on CS in KOA, and lack of reliable mean difference data for other outcomes were unavailable, additional outcomes were therefore exploratory, and the present study provides valuable preliminary data for future power calculations in this emerging research area.” (Page 9, Line 257–268)

Reviewer’s Comment: Body composition via bioelectrical impedance (BIA) is insufficiently sensitive for short-term changes and confounded by hydration status, especially when using creatine (which alters intracellular water).

Response: We appreciate this important observation. We fully acknowledge that BIA has certain limitations, particularly regarding sensitivity to hydration status, which can be influenced by creatine supplementation due to increased intracellular water retention. However, BIA remains a widely used, accessible, and non-invasive field method for estimating body composition, especially in low-resource research settings such as ours, where more advanced imaging modalities (e.g., DEXA, MRI) are not readily available.

To minimize variability, all measurements were standardized as participants were instructed to avoid food intake, exercise, and caffeine for at least 4 hours before assessment, and testing was performed at a similar time of day for all participants. (Page 9, Line 241–244)

We have also acknowledged this limitation in the Discussion section and recommend that future studies employing creatine supplementation use more precise techniques such as DEXA to validate these findings.

Isometric strength via sphygmomanometer is not a gold-standard measure; more reliable tools (isokinetic dynamometer) should be used or acknowledged as a limitation.

Response: We fully agree that an isokinetic dynamometer represents the gold standard for assessing isometric muscle strength. However, this study was conducted in a low-resource setting, where access to such advanced equipment is extremely limited. In fact, to our knowledge, there is only one isokinetic dynamometer available in the entire twin cities of Rawalpindi and Islamabad (capital of Pakistan), located at the Pakistan Sports Board, and not in any clinical or university-based rehabilitation center.

Given these limitations, the modified sphygmomanometer test (MST) was used as a practical and validated alternative. Previous studies have demonstrated high inter- and intra-rater reliability (ICC > 0.83) for both knee flexor and extensor measurements using MST, making it a suitable substitute in settings where dynamometry is not feasible.

We had already acknowledged this as a methodological limitation in the manuscript and highlighted the need for future studies in better-equipped settings to employ isokinetic dynamometry for more precise quantification, as can be seen in discussion:

Lastly, in the current study hand held modified sphygmomanometer test and bioelectrical impedance analysis were used for quantifying muscle strength and body composition parameters, and it is suggested that future studies should incorporate more specialized tools such as isokinetic dynamometer, electromyography and magnetic resonance imaging (MRI) for quantifying muscle strength, performance and body composition. (Page 18 and 19, Line 497 to 504)

Reviewer Comment: Fall Risk Score instrument not identified — unclear validity or clinical interpretation.

Response: Fall risk was assessed using the Biodex Balance System (Biodex Medical Systems Inc., Shirley, NY, USA), which is shown to have a test–retest reliability ICC of 0.64–0.91 [40] and has been used previously in musculoskeletal conditions including KOA to evaluate balance impairments and fall risk [41-43]. For the fall risk assessment, par-ticipants were instructed to stand with their eyes open on an unstable platform set at stability level 6. Each participant completed three 20 second trials, the average of which was computed afterwards. Before the test, a familiarization trial was conducted to ensure participants understood the procedure, and those scores were not recorded. During the test, participants were instructed to maintain their balance. A higher score on the Biodex Balance System is indicative of a greater risk of falling [42, 44]. (Page 9, Line 246 to 255)

Reviewer Comment: The analysis relies heavily on repeated t-tests and mixed ANOVA without correction for multiple outcomes or checking assumptions (sphericity, homogeneity).

Response: We appreciate the reviewer’s concern. We would like to clarify that the statistical assumptions and appropriate corrections were, in fact, observed and applied during the analysis. The normality of data was verified using graphical methods (boxplots and Q–Q plots), and homogeneity of variances was assessed prior to applying parametric tests. Between-group and within-group comparisons were conducted using independent and paired t-tests, respectively, only after confirming that data met these assumptions.

For the main inferential analysis, a mixed ANOVA was employed to evaluate the interaction effect (group × time), main effect of time, and main effect of treatment group. To address the reviewer’s concern about multiple testing, Bonferroni correction was applied to control for Type I error inflation. Furthermore, partial eta squared (η²p) values were reported to quantify effect sizes.

All analyses were performed using SPSS v21.0, with a 95% confidence interval and p < 0.05 considered statistically significant. This information has now been clarified and emphasized in the revised Methods → Statistical Analysis section to ensure transparency.

All of this information can be found in the manuscript as well on Page as shown below:

“Normality of the data was checked using graphical method (boxplot and QQ-plot). Descriptive statistics were reported in the form of Mean ± Standard Deviation (S.D) and Mean Differences were reported with 95% confidence intervals since the data were normally distributed. Fischer exact test was used to compare the two groups in terms of categorical variables i.e. gender and grade of knee OA. Independent T-test and paired- T-test were used for between group and within group comparisons. Furthermore, Mixed ANOVA was used to compare the two groups with Bonferroni correction to determine the interaction effect (treatment group x time), main effect (time) and main effect (treatment group).  In addition, partial eta squared (η²p) was reported as a measure of effect size to interpret the magnitude of these effects. All the statistical analysis was carried out using SPSS v 21.0 and a confidence interval of 95% was used, with a p-value of less than 0.05 considered significant.” (Page 9 and 10, Line 269 to 280)

Reviewer comment: The effect size reporting (η²p) is appropriate but not consistently interpreted (e.g., small/moderate/large).

Response: We appreciate the reviewer’s observation. The effect sizes (η²p) were reported consistently across all outcomes; however, we intentionally chose not to categorize them as small, moderate, or large, since interpretation can vary depending on clinical and contextual perspectives. What may be considered a “large” effect statistically might represent only a modest clinical change, and vice versa. Therefore, to avoid imposing a potentially subjective interpretation, we preferred to present the raw effect size values, allowing readers and clinicians to interpret their practical significance in light of their own clinical experience and judgment.

Reviewer comment: Confidence intervals are sometimes misused — negative signs for differences are confusing; precision intervals overlap substantially, suggesting possible overinterpretation.

Response: We appreciate the reviewer’s observation. The presence of negative signs in the confidence intervals was intentional, as they simply indicate the direction of change between pre- and post-intervention or between-group comparisons. In instances where the post-intervention mean was higher than the baseline (or vice versa), the resulting difference could naturally yield a negative value depending on the subtraction order (e.g., pre–post or group A–group B). These signs therefore do not represent an error or misuse but rather convey the directionality of the observed change. 

Reviewer comment: No intention-to-treat (ITT) analysis reported — necessary for RCT validity, even if no dropouts occurred.

Response: We appreciate the reviewer’s point regarding ITT analysis. In our trial, no participant dropouts or protocol deviations occurred; all 40 participants completed the intervention and post-assessments as per protocol. Therefore, the per-protocol analysis and ITT analysis would have produced identical results. However, we have now clarified this in the Methods section for transparency, stating that “an intention-to-treat approach was considered; however, as no dropouts occurred, analyses were conducted on the full dataset of all randomized participants.” (Page 10, Line 281 to 283)

Reviewer Comment: The manuscript repeatedly claims “significant improvements” in multiple variables, but the magnitude of these improvements is statistically small and clinically negligible.

Response: We appreciate the reviewer’s observation. In our manuscript, we reported statistically significant findings based on the conventional threshold of p < 0.05. However, we did not intend to overstate their clinical importance. We agree that statistical significance does not necessarily imply clinical relevance, and we have been careful to highlight this distinction in the Discussion section. Wherever possible, we have compared our results against established minimal clinically important difference (MCID) values for knee osteoarthritis to clarify that some statistically significant changes may not be clinically meaningful. We have also revised the wording throughout the manuscript to ensure that statistical and clinical interpretations are clearly differentiated. 

Reviewer Comment: The discussion acknowledges the lack of minimal clinically important difference (MCID) in VAS, but other non-significant findings are still framed as positive — indicating interpretive bias.

Response: We appreciate the reviewer’s feedback. Our intention was to report and interpret the statistically significant results objectively (i.e., p < 0.05) without overstating their clinical importance. We acknowledge that MCID values were not available in the literature for all outcome variables assessed in this study. Therefore, while statistical significance was reported where applicable, clinical significance could only be discussed for variables with established MCID thresholds (e.g., VAS).

Reviewer Comment: The short duration (4 weeks) cannot support conclusions about muscle hypertrophy or body composition.

Response: We acknowledge it is a big limitation of our study but was unavoidable due to lack of funding and patient compliance. We agree with the reviewer that a 4-week duration may not be sufficient to observe substantial muscle hypertrophy or major changes in body composition. However, the study did not aim to assess long-term hypertrophic adaptations but rather to explore early physiological responses to creatine supplementation and exercise therapy in knee osteoarthritis. Given the constraints of conducting research in a low-resource setting and the participants’ limited availability for longer follow-ups, a 4-week period was chosen to ensure compliance and completion. We have already clarified this in the Discussion section and acknowledged that longer-term studies are warranted to confirm the sustainability and magnitude of these effects, as mentioned in text:

It is also imperative to point out that the duration of treatment and follow up of 4 weeks in the current study was very short in terms of truly analyzing the effects on muscle size, as it may take 8 – 12 weeks to undergo structural changes in muscles, and the initial increase in muscle strength has a major contribution from neural factors and increased neural drive rather than increase in muscle size [56]. Similarly, the percentage increase in lower extremity lean mass can also be partially attributed to increase in ICW ratio. This relatively short duration of the intervention (four weeks) may be viewed as a limitation, as longer trials (8–12 weeks) are generally required to observe substantial hypertrophic or long-term functional changes in muscles. However, this study aimed to examine early responses and feasibility in a low-resource context where lack of funding, participant adherence, and clinical expectations favor shorter interventions. Unfortunately, conducting a longer follow-up was not feasible due to resource constraints and limited participant compliance, as the research was carried out in a low-resource setting (Pakistan) and was mostly self-funded. Moreover, patients in this context often seek short-term symptomatic relief to quickly return to work and daily responsibilities, making prolonged follow-up challenging. Nonetheless, we agree that assessing the long-term durability and retention of benefits is crucial. This study therefore provides preliminary data and serves as a foundation for future research to evaluate sustained outcomes through extended follow-up durations. (Page 17 and 18, Line 436–454)

Reviewer comment: ROM improvements are likely attributable to therapy rather than creatine; yet the text implies causal contribution.

Response: We respectfully disagree with the reviewer’s interpretation. We have not implied that creatine supplementation independently improved ROM. In fact, as explicitly stated in the Discussion section, improvements in ROM were attributed to the effects of physical therapy and exercise, and we specifically noted that creatine supplementation had no additive effect on ROM outcomes. We have rechecked the manuscript to ensure this interpretation is clearly conveyed, as shown below:

“In terms of range of motion both the treatment groups showed statistically significant improvements, however no significant differences were observed between the two groups, signifying that the improvement in knee range of motion was due to joint mobilization and exercise and had no additive benefit from CS.” (Page 16 and 17, Line 398–402)

Reviewer comment: The authors cite literature (e.g., Neves et al., Peeler et al.) selectively, ignoring trials with null or conflicting results in older adults and OA populations.

Response: We respectfully disagree with this comment. The studies by Neves et al. and Peeler et al. actually include findings that align with and contrast our own, and both have been discussed appropriately in our discussion section to provide a balanced interpretation. The evidence base regarding creatine supplementation in individuals with knee osteoarthritis (KOA) remains limited, and we have included all available and relevant trials in this specific population to the best of our knowledge.

Our focus was specifically on individuals with KOA, not the general older adult population, hence studies conducted exclusively in healthy older adults were not prioritized. Nevertheless, we acknowledge broader literature in older adults, such as the recent systematic review and meta-analysis by Sharifian et al. (2025), which also supports the positive influence of creatine supplementation in this group. However, as mentioned, this population differs from our study cohort and therefore was not central to our evidence selection strategy.

Sharifian G, Aseminia P, Heidary D, Esformes JI. Impact of creatine supplementation and exercise training in older adults: a systematic review and meta-analysis. European Review of Aging and Physical Activity. 2025 Dec;22(1):1-4.

Reviewer Comment: Figures and Tables: Contain excessive decimals and unnecessary precision (e.g., mean difference [−39.70; −35.96]).

Response: All numerical values have been reported up to two decimal places, which aligns with standard scientific reporting practices for continuous data. This level of precision ensures clarity and consistency in data presentation without overstating accuracy. Exceptionally low p-values have been reported as p < 0.001, which is also standard practice in scientific manuscripts. Therefore, we believe the current level of precision in our figures and tables is appropriate and consistent with established research reporting guidelines.

Reviewer Comment: Ethical statement: Appropriate, but the inclusion of “AI tools such as ChatGPT and Grammarly” should be checked against the journal’s disclosure policy (many high-impact journals require detailed AI use justification or disclaim it entirely).

Response: The inclusion of AI tools has already been acknowledged

Reviewer Comment: References: Many self-citations (Osama et al.) and limited inclusion of high-quality RCTs or meta-analyses (e.g., no reference to latest 2023/2024 OA management guidelines or large creatine meta-analyses). Furthermore, you should cite the following articles: 10.3390/sports11040091; 10.3390/jpm14080784.

Response: We appreciate the reviewer’s feedback regarding citations. To clarify, there are currently no large-scale meta-analyses specifically investigating creatine supplementation in knee osteoarthritis populations. The only comprehensive review directly addressing this topic is indeed by Osama et al., which was cited as it is the most relevant and contextually appropriate source available, as shown below:

Osama, Muhammad, Bruno Bonnechère, Jean Mapinduzi, and Farooq Azam Rathore. "Synergistic Effects of Creatine Supplementation and Resistance Training in the Management of Knee Osteoarthritis: A Narrative Review." Journal of Pakistan Medical Association (JPMA) 75, no. 10 (2025): 1654-58.

Regarding the suggested articles (10.3390/sports11040091; 10.3390/jpm14080784), while they focus on broader management perspective in knee osteoarthritis, rather than effects of creatine supplementation in knee osteoarthritis specifically, as shown below however we have included one of them in the revised version to acknowledge related work in the field.

Tarantino D, Theysmans T, Mottola R, Verbrugghe J. High-intensity training for knee osteoarthritis: a narrative review. Sports. 2023 Apr 20;11(4):91.

Tarantino D, Forte AM, Picone A, Sirico F, Ruosi C. The effectiveness of a single hyaluronic acid injection in improving symptoms and muscular strength in patients with knee osteoarthritis: a multicenter, retrospective study. Journal of Personalized Medicine. 2024 Jul 24;14(8):784.

Our intention is to maintain scientific rigor and relevance rather than self-citation bias.

Reviewer comment: Eligibility criteria lack clarity on baseline physical activity, supplement use, and comorbidities (e.g., diabetes, cardiovascular disease).

Response: We appreciate the reviewer’s observation. Our eligibility criteria were primarily focused on clinical and radiographic characteristics of knee osteoarthritis to ensure a homogeneous KOA population in terms of disease severity. As stated in the manuscript, patients aged 40–70 years with radiologically confirmed KOA of Kellgren grade III or less were included, while those with serious pathology, inflammatory disorders, trauma, fractures, radiculopathy, myelopathy, recent supplement usage or intra-articular steroid therapy were excluded.

Baseline physical activity levels, and comorbidities such as diabetes or cardiovascular disease were not used as exclusion criteria, as the study aimed to represent real-world patients typically encountered in low-resource rehabilitation settings. All participants were medically screened by a physiatrist before enrollment to ensure safety for exercise and supplementation. Nonetheless, we acknowledge the lack of detailed control for these factors as a limitation

Reviewer comment: Ethical approval is appropriately reported but lacks a statement on data monitoring or adverse event reporting.

Response: Ethical approval was taken from Advance Study and Research Committee, Isra Institute of Rehabilitation Sciences (Ref#1809-PhD-004) and Foundation University Islamabad (Ref#FF/FUMC/215-30/Phy/20), and the study was prospectively registered at clinicaltrials.gov (NCT04423887). Informed and written consent was acquired from all participants, and participants had the right to withdraw from the study at any point. As the study involved a short-term, low risk intervention consisting of supervised physical therapy and CS within safe, evidence based dosages, no formal data monitoring committee was formed. Participants were, however, monitored throughout the intervention period for any adverse events.  It is important to note however, that no adverse events were reported. (Page 3, Line 100 to 109)

Reviewer comment: The inclusion of both mean differences and p-values without visual confidence intervals reduces interpretability

Response: Confidence intervals are included with mean difference in square brackets.

Reviewer Comment: Graphical presentation (Figures 3–4) lacks error bars and individual data points.

Response: Individual data points and error bars were deliberately omitted from Figures 3–4 to maintain visual clarity and prevent overcrowding, as the same detailed numerical information including means, standard deviations, and confidence intervals is comprehensively presented in the corresponding tables. The figures were designed to provide a clear visual summary rather than duplicate tabular data.

Reviewer Comment: No adverse events or safety monitoring data reported, despite creatine’s known effects on renal parameters and water balance.

Response: No adverse effects were reported in the study. Moreover, as already mentioned and discussed in the discussion:

“Regarding the potential adverse effects of CS, research indicates that it may lead to weight gain; however, this increase is attributed to muscle mass rather than fat ac-cumulation [29]. Additionally, CS has been found to enhance water retention within muscles, leading to an ICW, which may also contribute to a rise in overall body mass [30]. This rise in ICW has led to the assumption that CS causes water retention. How-ever, creatine itself is an osmotically active compound, resulting in a short-term increase in ICW rather than generalized water retention [30]. This acts as a stimulus for muscle growth and is linked to increased strength and improved functional capacity [54]. Over the long term, increase in total body water relative to muscle mass does not occur, indicating that CS may not lead to water retention [30]. In fact, a meta-analysis by Forbes SC et al. (2019) demonstrated that CS, when combined with resistance training, contributes to a reduction in fat mass in older adults [58]. Anecdotal reports suggest that CS may cause side effects such as kidney damage; however, no scientific evidence has been found to support these claims [27, 30]. Additionally, studies have shown no sig-nificant effects of CS on renal or hepatic toxicity [59-61]. A meta-analysis published in 2019 found that CS does not cause renal damage, as it does not lead to significant changes in serum creatinine or plasma urea levels [60]. Moreover, in the context of KOA, Neves et al. reported no significant difference in creatinine clearance between the CS group and the placebo group [59]. Thus, based on the available literature, CS can be considered safe with no significant evidence of renal or hepatic toxicity”. (Page 18, line 469 to 491)

No registration summary from ClinicalTrials.gov included (should match NCT04423887 record).

Response: Clinical trial registration number already mentioned

CONSORT checklist and flow diagram (Figure 3) incomplete — no numbers for assessed, excluded, or analyzed participants.

Response: All aforementioned concerns are already mentioned in Figure 1.

Reviewer comment: No cost-benefit analysis or subgroup analyses (e.g., by gender, age, KOA grade).

Response: As presented in Tables 1 and 2, both groups were already compared for baseline characteristics including gender, age, KOA grade, weight, height, and BMI, with no significant between-group differences observed. Therefore, additional subgroup analyses were not required. Furthermore, a cost-benefit analysis was beyond the scope and objectives of the present study, which focused primarily on the physiological and functional effects of creatine supplementation in individuals with knee osteoarthritis.

Reviewer comment: The conclusion overstates the results. 

Response: the conclusion has been revised and is now as follows:

Physical therapy and resistance exercise training play a crucial role in the man-agement of KOA by improving pain, function, range of motion, muscle strength, and body composition. However, creatine supplementation has the potential to offer ad-ditive benefits, particularly in improving muscle strength, skeletal muscle mass, in-tracellular water ratio, and functional performance. These findings suggest that crea-tine supplementation can act as a safe and potentially valuable adjunct to conventional physical therapy, primarily when combined with resistance exercise. However, in the absence of exercise, its benefits may be limited. Longer duration studies with larger samples are needed to confirm clinical relevance, assess long term effects, and evaluate cost effectiveness. Future studies should also consider advanced assessment tools, such as isokinetic dynamometry, electromyography, and MRI, to more precisely quantify muscle strength, function, and body composition. (page 19, line 506 to 517)

Round 2

Reviewer 1 Report

Comments and Suggestions for Authors

The authors largely improved the manuscript and answered all the questions.

I think the manuscript is now well-written and provides a clear and detailed description of the study.

One comment –

In the abstract in the results part, you describe that KOOS was improved better in the CR group, yet in the conclusion, you state that QOL did not improve. This is confusing, as KOOS is your measurement for QOL. You should state in the results part as well the QOL part has no interaction effect.

Author Response

We would like to thank the respected editorial board member and the respected reviewers involved in the process of reviewing this manuscript and providing their kind feedback for the purpose of improving the quality of this manuscript.

Reviewer's Comment: In the abstract in the results part, you describe that KOOS was improved better in the CR group, yet in the conclusion, you state that QOL did not improve. This is confusing, as KOOS is your measurement for QOL. You should state in the results part as well the QOL part has no interaction effect.

Response: No significant differences were observed between the two groups at baseline. After 4 weeks of treatment a significant interaction effect (treatment group x time) was observed for VAS (p=0.001), fall risk score (p<0.001), KOOS overall score (p<0.001), IMS (p<0.001) and body composition parameters (p<0.05) in favor of the CS group. However, no significant interaction effect was observed for knee ROM and KOOS QOL subscale. (Page 1, Line 28 to 33)

Reviewer 2 Report

Comments and Suggestions for Authors

Thank you for taking the time to implement the changes and address my comments. I fully accept them and congratulate you on creating a high-quality manuscript.

Author Response

We would like to thank the respected editorial board members and the respected reviewers involved in the process of reviewing this manuscript and providing their insights for the purpose of improving the quality of this manuscript.

Reviewer 3 Report

Comments and Suggestions for Authors

Dear Authors,

I am satisfied with the revised version of the manuscript and with the responses to the issues I raised. Good job.

Author Response

(The authors gave the same response as above.)
